# Development and Pharmacokinetics of a Novel Acetylsalicylic Acid Dry Powder for Pulmonary Administration

**DOI:** 10.3390/pharmaceutics14122819

**Published:** 2022-12-15

**Authors:** Adam Pacławski, Stavros Politis, Evangelos Balafas, Ekaterini Mina, Paraskevi Papakyriakopoulou, Eirini Christodoulou, Nikolaos Kostomitsopoulos, Dimitrios M. Rekkas, Georgia Valsami, Stefano Giovagnoli

**Affiliations:** 1Department of Pharmaceutical Technology and Biopharmaceutics, Jagiellonian University Medical College, Medyczna 9 St., 30-688 Cracow, Poland; 2Section of Pharmaceutical Technology, Department of Pharmacy, National & Kapodistrian University of Athens, 15784 Athens, Greece; 3Center for Clinical, Experimental Surgery and Translational Research, Biomedical Research Foundation of the Academy of Athens, 11527 Athens, Greece; 4Department of Pharmaceutical Sciences, University of Perugia, Via del Liceo 1, 06123 Perugia, Italy

**Keywords:** acetylsalycilic acid, aspirin, dry powders, inhalation, pulmonary delivery, pharmacokinetics

## Abstract

Aspirin is an historic blockbuster product, and it has been proposed in a wide range of formulas. Due to exacerbation risks, the pulmonary route has been seldom considered as an alternative to conventional treatments. Only recently, owing to overt advantages, inhalable acetylsalicylic acid dry powders (ASA DPI) began to be considered as an option. In this work, we developed a novel highly performing inhalable ASA DPI using a nano spray-drying technique and leucine as an excipient and evaluated its pharmacokinetics compared with oral administration. The formulation obtained showed remarkable respirability and quality features. Serum and lung ASA DPI profiles showed faster presentation in blood and higher retention compared with oral administration. The dry powder was superior to the DPI suspension. The relative bioavailability in serum and lungs claimed superiority of ASA DPI over oral administration, notwithstanding a fourfold lower pulmonary dose. The obtained ASA DPI formulation shows promising features for the treatment of inflammatory and infectious lung pathologies.

## 1. Introduction

Acetylsalicylic acid (ASA) is a well-known blockbuster analgesic, antipyretic, and anti-inflammatory drug. The long tradition of ASA-based medicines has led to a number of marketed products, most of which come as oral preparations [1]. The oral route is preferred for ASA administration, being generally low cost, highly compliant, and well-established from a regulatory and industrial point of view. However, this administration modality may not always be as effective when localized action is desirable. This can be the case for local inflammation events which occur as a result of infections or other local or systemic diseases. In this regard, lung diseases and infections could be treated more efficiently through the pulmonary route. Inhalation is recognized as one effective strategy to enhance local drug action [2], thus reducing systemic exposure and dose, which could be convenient, especially to limit off target effects. Moreover, the lungs can also represent a port of entry to the systemic circulation, owing to the large surface area and blood perfusion that favor rapid drug absorption. Such features have recently encouraged the exploration of this delivery approach for anti-inflammatory drugs, such as ibuprofen, indomethacin, and diclofenac, even in the form of nanoparticles [3,4,5,6,7].

Reports have shown potential advantages of ASA in tuberculosis or other lung infections as supporting treatment. Byenr et al. [8] found that ASA can enhance the effect of pyrazinamide in the treatment of tuberculosis in a mouse model and pointed to the need for further experiments to check possibilities of clinical implications of ASA in tuberculosis treatment in humans. Previous experiments showed that ASA can reduce several genes involved in transcription, translation, and energy production in Mycobacterium tuberculosis [8]. Pulmonary formulation of ASA can reduce the required dose of the drug, which limits the known concerns for ASA respiratory exacerbations [9]. On the other hand, while it should be avoided in individuals with aspirin sensitivity history, low dose aspirin safety in COPD and asthma conditions has been postulated in the clinical literature. In fact, recent works have shown that COPD conditions can be improved by daily aspirin intake, [10] and patients with acute exacerbation COPD benefited from aspirin administration with reduced mortality, invasive mechanical ventilation, and hospitalization time [11]. Moreover, aspirin demonstrated a potential protective role in patients with acute respiratory disease syndrome (ARDS) [12], and low-dose aspirin did not alter inflammatory markers and lung function in patients with mild-to-moderate persistent asthma [13]. Ten years of regular aspirin use was even associated with a more than 50% reduction in the rate of emphysema progression [14]. When inhaled, aspirin–lysine was found safe and easily controlled, compared with oral challenge, and a repeatable procedure not associated with any late-phase bronchoconstriction or increase in nonspecific airway responsiveness [15]. Recently, aerosolized LASAG (lysine–ASA–glycine, BAY 81-8781), a product licensed for aspirin IV medication showed superior anti-influenza efficacy compared with oral administration in animal models and hospitalized patients [16]. The application of inhaled aspirin has been postulated as an adjunctive therapeutic option in COVID-19 as well [17].

Therefore, ASA pulmonary delivery, albeit cautiously approached, is supported by a solid rationale for the treatment of lung inflammation, even when associated with infections and even for chronic systemic therapies. In this regard, a recent phase 1, pilot open-label, single-dose-escalation trial showed the superior efficacy of pulmonary administration of ASPRIHALE^®^, an under-development aspirin DPI formulation, for antiplatelet therapy [18]. To the best of our knowledge, this is the only report on an inhalable aspirin dry powder product specifically developed for lung delivery. In such a context, the development of well-designed and highly performing delivery strategies is fundamental to warrant the desired treatment efficacy and safety profiles. For molecules such as ASA, liquid forms are not recommended due to the fast hydrolysis of the drug when exposed to aqueous environments. On the other hand, dry powders can suit the ASA need for superior stability and penetration into the respirable tract of the lungs.

Inhalable dry powders are produced mainly by spray-drying, a technique that is broadly described in the literature and industrially established, as it allows one-step fabrication of particles with controlled features. An advanced version of the conventional spray-drying technique is the high performance (HP) nano spray-drying process developed in 2009 by the Büchi company. This approach relies on a piezo-electric atomizing mesh and an electrostatic particle collector that enables the production of nanosized particles at high yields and lower spray temperatures [19]. The literature on the use of such a technique has grown over the years, and several works have described the successful production of inhalable particles for the treatment of inflammation in cystic fibrosis [20], lung fungal infections [21,22], tuberculosis [23], and hypertension [24].

Therefore, in this work, we employed the HP nano spray-drying technique to produce a novel high performing ASA inhalable powder. After assessing formulation properties and behavior, pharmacokinetics (PK) of the inhaled formulation was compared with conventional oral administration. The obtained ASA inhalable powder shows promising features that, beyond the above anticipated antiplatelet effect, hold promise for a potential application in the treatment of lung inflammatory and infectious diseases.

## 2. Materials and Methods

### 2.1. Respirable ASA–Leu Dry Powder Preparation

ASA–leucine dry powders were prepared by using the Buchi nano spray-dryer B90 (Büchi, Milan, Italy). Sodium acetylsalicylate and D-leucine (Sigma Aldrich, Milan, Italy) were dissolved in water/absolute ethanol (Sigma Aldrich, Milan, Italy) solution, mixed thoroughly and immediately spray-dried. The method was characterized by applying the experimental design technique described here below.

### 2.2. Experimental Design

A 2^6−2^ fractional factorial design was employed to assess the effects of the above process and formulation parameters on the Critical Quality Attributes (CQAs) of the new inhalable dry powder.

Mathematical models were elaborated by using Design Expert^®^ v. 8.0.7.1 (Stat-Ease, Inc., Minneapolis, MN, USA). Experimental data were fitted by a general quadratic model (Equation (1)).
(1)$$y=b_{0}+\sum_{i=1}^{k}b_{i}x_{i}+\sum_{i=1}^{k}\sum_{j=1}^{k}b_{ij}x_{i}x_{j}+\sum_{i=1}^{k}b_{ii}x_{i}^{2}+e$$
where *y* is the response, *b_0_*, *b_i_*, *b_ii_*, and *b_ij_* are the intercept, linear, quadratic and interaction regression coefficients of the *x*_i_, *x_j_* level of the *i*-th, *j*-th factor, respectively, and *e* is the residual random error of the model. The levels of the selected input parameters were based on preliminary experiments and considerations on method technical limitations. Experiments were conducted in a randomized manner to reduce bias.

Factors and responses of the design are reported in Table 1. A 4.0 µm pore size stainless steel membrane and the maximum spray frequency of 60 kHz were chosen to nebulize the solutions. Other process parameters were inlet temperature, air flow rate, and drying chamber pressure. Formulation parameters were total solute concentration, ASA/Leu ratio, and ethanol percentage (Table 1). Process yield, percent emitted dose (ED), fine particle fraction (FPF), and ASA content were selected as responses.

ANOVA analysis was performed for model statistical assessment and the corresponding response surfaces were built. The surface plots related to each response were combined into a desirability plot (Equation (2)) to delimit the region of the design space where the user-defined criteria profiling system target properties are met.
(2)$$f\left(d_{\left(g\right)}\right)=\sum_{i=1}^{M}w_{i}d_{i}/\sum_{i=1}^{M}w_{i}$$
where *f(d(g))* is the general desirability, *di* is the desirability of the *i*-th response, *M* is the number of response variables, and w is the user-specified weight.

Target properties of the obtained ASA powders were established as follows: process yield ≥ 50%, ED ≥ 90%, FPF ≥ 50%, and ASA content ≥ 40%, giving higher priority to FPF and ED criteria satisfaction over the other responses.

### 2.3. ASA Quantification

ASA–leucine dry powders were analyzed to determine the actual amount of ASA in the particles. The amount of ASA was evaluated by UV spectrophotometry using a UV/VIS Agilent 8453 Spectrophotometer (Agilent, Germany). A calibration curve of ASA was built in absolute ethanol solution in the concentration range 3–18 μg/mL (r2 = 0.99422). Precisely weighed amounts of ASA–Leu particles were dissolved in a volume of absolute ethanol solution and analyzed at λ_max_ = 225 nm upon proper dilution. The percentage of ASA in the powder was expressed according to the following equation:(3)$$\mathrm{ASA}=\frac{\mathrm{Amount}\;\mathrm{of}\;\mathrm{ASA}\;\mathrm{measured}}{\mathrm{Amount}\;\mathrm{of}\;\mathrm{powder}\;\mathrm{weighed}}\times 100\%$$

All measurements were performed in triplicate and the results expressed as a mean (SD).

### 2.4. Morphology and Particle Size

Powder morphological analysis was conducted by Scanning Electron Microscopy (SEM) using a FEG LEO 1525 microscope equipped with a Bruker Energy Dispersive X-rays (EDX) system. The acceleration potential voltage was between 2 and 10 keV. Samples were directly placed onto carbon-tape-coated aluminum stubs. The stubs were sputter coated with chromium at 20 mA for 30 s, prior to imaging.

ASA–leucine particles were analyzed for size by image analysis using the ImageJ software. At least one thousand counts were performed for each picture. Particle size was determined calculating the Feret’s diameter (FD) over a batch of three images. The population size distribution was then evaluated by calculating the Feret’s mean diameters (FMD) of the population and the S.D. of the population distribution.

### 2.5. Powder Flowability and Yield

The tapped density (ρt) of the MP was determined in triplicate samples by an ERWEKA tapped density tester SVM102 (Heusenstamm, Germany). Measurements were carried out by gravity filling of a 5 (± 0.05) mL cylinder with a known mass of powder, followed by a 2500 taps cycle. Tapped density and the freely settled bulk density (ρB) were calculated according to Equation (4).
(4)$$\rho_{t}=\frac{\mathrm{weight}\;\mathrm{of}\;\mathrm{powder}}{\mathrm{tapped}\;\mathrm{volume}}$$

The measured ρ*_t_* and ρ*_B_* values were used to calculate the Hausner’s ratio (H) (Equation (5)), for which H = 1.25 represents the upper flowability limit.
(5)$$H=\frac{\rho_{t}}{\rho_{B}}$$

The process yield was calculated by dividing the powder %*w*/*w* collected in the product collection chamber by the total amount fed to the instrument.

### 2.6. Thermal Analysis

Differential scanning calorimetry (DSC) analysis was performed on dry powder samples under nitrogen atmosphere by using a Mettler Toledo DSC 821 calorimeter (Mettler Toledo, Milan, Italy) equipped with a liquid nitrogen cooling system. Samples were accurately weighed and transferred to a 40 μL aluminum pan and sealed with a holed pan lid. Scan was performed between 25 and 200 °C or 350 °C at 10 °C/min heating rate. Pure drug and excipients were analyzed for comparison.

### 2.7. Aerodynamic Assessment

Aerodynamic assessment of the obtained dry powders was performed using a twin-stage glass impinger (TSGI) (Disa, Milan, Italy). As per standard protocol, the first stage of the TSGI was loaded with 7 mL and the second stage with 30 mL of a 1:1 water/ethanol solution. Capsule filling was carried out manually. After turning on the aspiration set at 60 ± 5 L/min, the capsule into the Handi-Haler was perforated and the aspiration was maintained for 5 s following the European Pharmacopoeia (apparatus A) guidelines [25]. Three capsules Vcaps1 number 3 were used for each experiment. Samples were collected and analyzed by the method reported above. The fine particle fraction (FPF) and fine particle dose (FPD) were calculated as percentages of the emitted dose (ED). The ED was expressed as a percentage of the nominal dose. All determinations were performed on triplicate samples.

### 2.8. Pharmacokinetic Study

#### 2.8.1. Animal Model

Healthy male Wistar (RccHan:WIST, ENVIGO) rats, 16 weeks old and mean body weight 513 ± 43 g, were used in the present study. Animals were housed in individually ventilated cages (Tecniplast, Varese, Italy) under specific pathogen-free (SPF) constant environmental conditions (12 h light:dark cycle), temperature 22 ± 2 °C, and relative humidity 45 ± 10%). The animals were fed with irradiated pellets (4RF22, top certified diet, Mucedola Slr, Italy) and had access to food and tap water on demand. The cage bedding comprised corncob granules (REHOFIX^®^, J. Rettenmaier & Söhne Co., Rosenberg, Germany); the cages and their bedding were changed weekly. Regular screening according to a health-monitoring program was conducted in compliance with the Federation of European Laboratory Animal Science Associations’ recommendations. The study was approved by the Veterinary Authorities of Region of Athens, Greece (Ref. number 3371/05-07-2018).

#### 2.8.2. Administration and Sampling Protocol

Animals were randomly divided into 3 groups (n = 5). One group was administered orally by gavage with an ASA-saturated solution, at a dose of 7.0 mg/kg, corresponding to a human equivalent dose of 80 mg for a 70 kg average body weight [26]. Two groups were treated by pulmonary administration of ASA dry powder (ASA DPI) at one-fourth of the oral dose, corresponding roughly to 1.8 mg/kg, adopting two modalities: (i) intrapulmonary instillation of 50 μL ASA DPI suspension (2.15 ± 0.47 mg ASA DPI) in normal saline, accurately measured and pipetted with a Gilson Pipetman M100, (5–100 µL ± 0.35–0.40 µL) pipette through a tracheal catheter and (ii) pulmonary insufflation of ASA DPI (1.43 ± 0.7 mg) by the PenWu Device dry powder insufflator, BJ-PW-FM-R (Bio Jane Trading Limited, Shanghai, China).

Blood and lung samples were taken at 10, 15, 30, 60, 120, 240, and 360 min. Blood samples were transferred into non-heparinized Eppendorfs and centrifuged at 10,000 rpm for 10 min to obtain serum. For blood sampling, animals were anesthetized using an inhalation anesthetic agent (Isoflurane; Iso-Vet, Piramal Enterprises ltd, India), and blood was collected via cardiopuncture following cervical dislocation. Lung samples were collected thereafter and thoroughly rinsed with normal saline. Serum and lung samples were frozen at -80 ^o^C until HPLC analysis.

#### 2.8.3. Salicylic Acid Quantification

The quantification of salicylic acid (SA) and the active metabolite of ASA, in serum and in lung samples, was performed by an optimized high performance liquid chromatography (HPLC) method with fluorescence detection, using naproxen as internal standard (IS), based on the method of Hany et al. [27]. The HPLC system consisted of a system controller SCL-10AVP, a pump LC-20AD, an autosampler SIL-HTC, a fluorometric detector RF-20A, and a thermostated column compartment CTO-20AC (all from Shimadzu, Kyoto, Japan). Separation in serum was achieved by an ACE ^®^ C-18, Trimethysilane (TMS) 150 × 4.6 protected by a guard column at room temperature. The mobile phase consisted of either H_2_O 0.1% orthophosphoric acid (A) or ACN 0.1% TFA (B). A gradient profile was used for the chromatographic separation of the analyte and the IS as follows: from 0.01 to 3.50 min, 40% A and 60% B, flow rate 0.8 mL/min; from 3.50 to 5.50 min, 55% A and 45% B, flow rate 0.8 mL/min; from 5.50 to 9.00 min, 70% A and 30% B, flow rate 1.6 mL/min; from 9.00 to 9.01 min, 40% A and 60% B, flow rate 1.6 mL/min; from 9.01 to 10.00 min, 40% A and 60% B, flow rate 3.0 mL/min; and from 10.00 to 10.01 min, 40% A and 60% B, flow rate 0.8 mL/min. The running time of one sample was less than 15 min. The wavelengths of excitation and emission were set at 290 nm and 445 nm. The temperature of the autosampler was 4 °C, and the column temperature was 32 °C. An injection volume of 20 μL was applied for SA. Calibration curves were constructed using blank rat serum or blank rat lung homogenate spiked with various amounts of SA and standard amount of IS. The peak area proportions of SA to the IS versus SA concentrations were used for linear regression analysis.

A stock solution of SA was prepared in acetonitrile at a concentration of 1000 μg/mL, whereas a 1000 μg/mL stock solution of naproxen (IS) was prepared by dissolving the appropriate amount in methanol. All stock solutions were kept frozen at −20 °C. The working solutions of SA used for the calibration curves were freshly prepared using acetonitrile: 0.1% orthophosphoric acid (99:1 *v*/*v*) as diluent, at the time of analysis (concentration range of SA: 5.00–100 μg/mL). Working solution of IS was also prepared at a concentration of 28 μg/mL by dilution from stock solution with acetonitrile: 0.1% orthophosphoric acid (99:1 *v*/*v*) as diluent.

#### 2.8.4. Blood and Lung Sample Treatment

Serum sample preparation was carried out by vortex-mixing 100 μL of serum with 20 μL of IS working solution (naproxen 28 μg/mL dissolved in acetonitrile) and 280 μL of acetonitrile: 0.1% orthophosphoric acid (99:1 *v*/*v*) for protein precipitation. The obtained suspension was centrifuged at 10,000 rpm for 12 min at 5 °C, and 20 μL of the supernatant was immediately submitted to HPLC analysis. Using the same procedure, standard solutions from blank serum samples spiked with SA were prepared for calibration. For lung tissue analysis, excised lung tissue samples were homogenized with 3 mL of NaCl 0.9% *w/v* solution per mg of tissue. Then, by the same procedure described for the serum samples, 100 μL of tissue homogenate was submitted to analysis.

#### 2.8.5. Noncompartmental PK Analysis

Pharmacokinetic parameters after oral and pulmonary administration were calculated using sparse sampling NCA for data representing SA concentration vs. time for each administration route. The main calculated PK parameters are AUC_0-t_ (the area under the concentration–time curve from time 0 to the last experimental time point) and AUC_inf_ (the area under the concentration–time curve extrapolated to infinity), C_max_ and t_max_. PK parameters and their standard errors (SE) were calculated based on the mean concentration curve data combined with the available subject characteristic. The AUC_inf_ was calculated according to the following equation:(6)$${\mathrm{AUC}}_{\inf}={\mathrm{AUC}}_{0-t}+{\;\mathrm{AUC}}_{t-\infty }$$
where AUC_t-∞_ is the area under the concentration–time curve from time t to infinity, calculated by dividing the last concentration by the terminal slope λ, which was obtained by linear regression analysis on the last three or four points of the log-transformed concentration vs. time curve. The relative pulmonary vs. oral bioavailability (F) was calculated by the following equation:(7)$$F=\frac{{\mathrm{AUC}}_{\inf\left(\mathrm{pulm}\right)}\cdot \;{\mathrm{Dose}}_{\mathrm{oral}}}{{\mathrm{AUC}}_{\inf\left(\mathrm{oral}\right)}\cdot \;{\mathrm{Dose}}_{\mathrm{pulm}}}$$
where AUC_inf(pulm)_ and AUC_inf(oral)_ are the areas under the concentration—time curve from time 0 to infinity after pulmonary ASA DPI and oral ASA solution administration, respectively, while Dose_pulm_ and Dose_oral_ are the respective administered doses. Furthermore, the elimination half-life, *t*_1/2_, was determined as *t*_1/2_ = 0.693/λ after calculation of the terminal slope, λ (*k_el_*). The dose was expressed in nmol/kg of body weight. The AUC_% extrapolated_ was calculated according to Equation (8):(8)$${\mathrm{AUC}}_{\%\;\mathrm{extrapolated}}=\frac{{\mathrm{AUC}}_{\inf}\;-{\;\mathrm{AUC}}_{0-t}\;}{{\mathrm{AUC}}_{\inf}\;}*100$$

### 2.9. Statistical Analysis

Data were analyzed by Student’s *t* test at 95% and 99% significance level from at least three replicates.

## 3. Results and Discussion

### 3.1. Determination of Optimal Formulation Conditions

The nano spray-drying technique was chosen to produce ASA inhalable powders. This relatively recent spray-drying technology is still in its infancy but shows several advantages over conventional methods. Low temperatures, high yields, and the possibility to reduce the size of particles to nanoscale are among the major quotes. A significant body of literature attests to the good performances that can be ensured by this technique, even when dealing with difficult to treat materials, such as soft materials and biomolecules [28]. In our case, given that ASA is unstable in an aqueous environment, nano spray-drying was useful to limit temperature effects and increase the yield of the process.

A factorial design was employed to investigate critical parameters influencing the quality of the obtained inhalable ASA formulation. The complete experimental design layout is reported in Table 2. Statistical analysis and coefficient estimates of the obtained hierarchical mathematical models for the selected responses is reported in Appendix A. All responses but yield were in a quadratic relation with significant parameters as determined by ANOVA analysis. The parameters C, E, and F, namely the ethanol amount employed, air flow rate, and back pressure, respectively, resulted in being generally uninfluential, and therefore, the models were reduced accordingly by excluding the corresponding terms.

Noteworthy, ASA loading had a major impact on the properties of all preparations. In particular, the ASA content depended solely on drug loading (Appendix A, Figure 1b). Overall, when exceeding a certain threshold, ASA loading correlated with worsening of powder quality. Indeed, a higher ASA loading determined a higher cohesiveness that dramatically affected powder recovery. This clearly emerges from the inverse relation between yield and ASA loading highlighted by the surface plots (Figure 1a). Such a behavior can be easily understood from preliminary observations that evidenced the high cohesiveness and stickiness of ASA-alone spray-dried powders, which make the excipient-free approach unfeasible. In fact, direct spray-drying of 100% ASA powder led to a complete loss of the material that remained attached to the particle collector walls (not shown).

Coherently, FPF and ED also were inversely related to ASA loading (Figure 1c,d). Therefore, a higher ASA content was reflected in poorer aerodynamic performance of the obtained dry powders. In fact, satisfactory ED and FPF values were recorded below 60% *w*/*w* ASA loading that corresponded to approximately 50% *w*/*w* ASA content (Table 2, Figure 1b–d).

This detrimental effect of ASA content on particle features is clearly displayed in Figure 2, where the morphology of > 90% *w*/*w* ASA content particles is shown. The resulting powders were indeed irregular either in size or shape, with evident aggregation of submicrometric particles. Moreover, the presence of elongated drug crystals can be spotted in some of them.

Such crystals were ascribed to ASA which, at high loadings, can recrystallize upon droplet drying. The presence of large drug crystals obviously alters powder quality by increasing cohesiveness and promoting aggregation.

Theoretical flow properties were generally poor, as all runs except run# 5 and 20 showed partial or no flowability (H > 1.25) (Figure 3a). This was expected due to the small size of the obtained particles expressed as FMD (Figure 3b). Most preparations had a FMD below or close to 1 µm, with a large population of nanosized particles as proven by morphological observations, where the presence of few micronized particles and a vast majority of small irregular and buckled-shaped particles is displayed in most preparations (Appendix A). A small size favors particle adhesion due to a prevalent effect of attractive Van der Waals forces on the particle surface with respect to particle mass [29]. This correlates with the observed increase of H values with FMD (Appendix A). Such a behavior can hinder particle flow capacity and explains the poor H values in Figure 3a.

Tapped density was rather consistent for all runs, with values ranging between 0.15 and 0.27 g/cm^3^ (Figure 3a). Such low values of *ρt* further confirm the compactability of the obtained powders upon tapping cycles as a result of small particle size and adhesion.

However, the observed adhesion seemed not to impair powder emission from the DPI employed in this study. Except for those preparations with very high ASA content that, as already mentioned, showed poor aerodynamic properties, all other runs had from high to very high FPF and ED when compared with commercial standards. In fact, as shown in Figure 1c,d and Table 2, FPF and ED values exceeded 60% and 90% when ASA content was below 60% *w*/*w*. This is not surprising considering the documented properties of D-leucine as a glidant in inhaled formulations [30]. Overall, in line with the above comments, the conditions that ensured compliance with target features, corresponding to process yield ≥ 50%, ED ≥ 90%, FPF ≥ 50%, and ASA content ≥ 40%, were met when ASA loading and feedstock concentrations were between 50–55% *w*/*w* and 1–1.4% *w*/*v*, respectively, at the lowest temperature tested of 85 °C (Figure 4). This narrow optimality space warranted yield = 69%, ASA content = 42% *w*/*w*, FPF = 73%, and ED = 92%. Among all preparations, run# 7 matches such features the most (Table 2), and this formulation was then selected to be employed for the subsequent in vivo experiments. In fact, the formulation is the optimal compromise between high aerodynamic performances and good process and particle physical features.

Size distribution and morphology of the selected formulation are displayed in Figure 5. FMD was close to 1 µm and the particle population was characterized by micrometric and nanosized particles with a collapsed structure typical of spray-dried powders. Particles were mainly crystalline as shown by the presence of ASA melting transition upon thermal analysis (Appendix A).

These features met the goal to achieve a high-performing powder with a good drug load. This is particularly important to allow proper dosing in the lungs without delivering an excessive mass of material that would challenge the suitability of the inhaled ASA approach. Moreover, the extremely high FPF > 80% is, to the best of our knowledge, unprecedented in the literature. This amazing result is ascribable to optimized conditions and particle features that closely depend on leucine properties. Although not yet approved for lung delivery, leucine is a highly attractive excipient that will soon find full application in the inhaled medicines field [31]. In fact, leucine is fundamental in determining bulk particle behavior upon aerosolization by forming a shell on the particle surface that provides protection, thus preserving dry powder aerodynamic features [32,33]. Leucine has been employed to formulate inhaled antimicrobial [34,35], β2 adrenergic receptor agonists [36,37], corticosteroids [38], and statins [39] as well as to embed lipid or polymeric particles [40,41]. Moreover, clinical studies indicated a low local toxicity risk for inhaled leucine [42]. In particular, a leucine-containing DPI formulation was well-tolerated in both phase 1 and phase 2 trials [43,44] as well as in a 24-week phase 3 clinical trial [45].

Therefore, the proposed ASA DPI shows suitable features for pulmonary administration and is potentially highly translational.

### 3.2. Pharmacokinetic Study

On the basis of the above considerations, the run# 7 formulation was selected to investigate ASA PK. ASA in serum and lungs was evaluated after pulmonary administration of the prepared ASA DPI to Wistar rats either as suspension in normal saline or as a dry powder in comparison with oral administration. The oral dose was selected to correspond to a human equivalent dose of 80 mg for a 70 kg average body weight, while that for pulmonary administration was reduced to one-quarter of the oral dose. Such a dose was chosen considering, on the one side, the recognized capacity of inhaled products to produce higher lung drug levels with lower administered dose [46,47] and, on the other side to avoid toxicity issues. In fact, the conventional ASA oral dose is not completely translatable to lung delivery. DPI administration allows for the delivery of only a few hundred mg of powder to the lungs, owing to the intrinsic technical limitations of DPIs and MDIs and, most importantly, the risk for toxic exacerbations [48]. This is particularly relevant also considering that only approximately 40% *w/w* of the delivered powder is ASA, which increases the amount of powder required to grant the desired dose. Therefore, we selected a pulmonary dose one-fourth of the oral dose to prevent the above-mentioned issues.

Regarding ASA suspension, an average dose of 0.9 ± 0.14 mg of ASA was administered per animal, corresponding to 1.75 ± 0.27 mg/kg, while for ASA DPI, the actual pulmonary dose delivered to the animals was determined by assessing the emission from the device after each actuation that resulted 1.43 ± 0.7 mg, corresponding to 0.6 ± 0.3 mg (or 1.17 ± 0.59 mg/kg) of ASA per animal. The lower performance by dry powder insufflation may be explained by a slight powder clumping upon emission from the device in contact with fluids. In general, the actual dose was close enough to the nominal dose with acceptable reproducibility.

A customized method was validated for ASA analysis. In this regard, since ASA is hydrolyzed to SA in biological fluids, calibration was performed by plotting SA/IS peak area ratio vs. SA concentration. The method showed good linearity over the examined range (0.25–5.00 μg/mL), with fixed IS concentration at 1.40 μg/mL. An overall correlation coefficient of 0.9997 (±0.032% RSD) and 0.9995 ((±0.020% RSD) was obtained from six sets of calibration curves for serum and lung tissue, respectively. All calculated concentrations of the standard solutions and for the lower limit of quantification (LLOQ) were within ± 15% and ± 20% of the nominal value, respectively. In serum and lung homogenates, an acceptable 81% recovery was obtained (% RSD 2.8), while the biological matrix effect on the calibration curve was negligible. Six analytical runs of QC (quality control) samples were employed for method accuracy and precision assessment. The QC solutions were prepared in serum and lung homogenates with IS at 1.40 μg/mL at three different SA concentration levels: 0.25 μg/mL, 1.25 μg/mL, and 5.00 μg/mL. For each QC concentration level, within-run accuracy was ≤ 10.0% and < 15.0% for LLOQ, between-run accuracy was < 10%, within-run RSD was between 0.35% and 3.3%, and between-run RSD was <5%. In serum and lung homogenates, the LLOQ and limit of detection (LOD), determined measuring the signal at 3-fold and 10-fold the baseline noise, were 0.189 μg/mL and 0.236 μg/mL, and 0.062 μg/mL and 0.072 μg/mL, respectively.

The obtained concentration vs. time curves are displayed in Figure 6, while the estimated SA basic PK parameters after NCA are reported in Table 3 for serum and lungs. The profiles after pulmonary ASA DPI administration show a biphasic behavior both in serum and lungs, with a first smaller peak at 15 min and a second peak at 1 h. This behavior was explained by a promoted lung-to-blood diffusion of SA when inhaled in the dry powder form. Such an effect may be ascribed to an ASA DPI fine particle population that dissolves faster compared with larger particle populations. This hypothesis fits with the presence of submicron particles evidenced by SEM and particle size analyses (Figure 5). This interpretation is also supported by the absence of a biphasic behavior after administration of the suspension, where the fine powder population is already likely dissolved prior to insufflation (Figure 6). Moreover, the ASA DPI biphasic profiles in serum and lungs are qualitatively identical, which suggests an effect related to intrinsic powder characteristics rather than physiologic factors.

We chose to measure SA, as ASA-to-SA conversion in physiologic conditions is well characterized and, as also shown by our data, it occurs in a relatively short time frame. This hydrolytic process depends on specific environmental features as well as on the activity of specific enzymes in humans, such as Butyrylcholinesterase and a newly identified extracellular form of the platelet-activating Factor Acetylhydrolase 1b2 that account for aspirin hydrolysis in plasma [49]. Due to epigenetic and physiologic factors, enzyme activity varies among individuals, and this changes the rate of aspirin hydrolysis, which explains the clinically observed highly variable antiplatelet activity.

In our case, the time to reach SA peak levels seems coherent with the literature reporting serum *t_max_* between 0.5 and 1 h in rats depending on the modality of administration and dose [50,51]. However, the ASA DPI first peak at 15 min claims a faster partition of SA that may account for the enhancer effect of the nanometric particle subpopulation mentioned above. In this regard, the AUC_lung_/AUC_serum_ ratio was comparable for ASA DPI and the oral, while the ASA DPI suspension showed the highest serum partition, being already partially solubilized upon administration. Moreover, given that the ASA pulmonary dose was four times lower than the oral dose, particularly in serum, the oral AUC_last_ value was expected to be considerably larger than that after pulmonary administration. However, while the value related to ASA DPI suspension was nearly one-fourth compared with orally administered ASA, that of ASA DPI was only halved, and this gap was consistent both in serum or the lungs (Table 3). Moreover, the bioavailability of SA relative to oral was 133% and 97% in serum and lung, respectively, for the ASA DPI suspension and more than 300% for ASA DPI in both compartments (Table 3).

These results point to a higher efficiency and potential therapeutic advantage of delivering ASA directly to the lungs as a dry powder. Therefore, from the above observations, it can be inferred that pulmonary administration expedites SA presentation into blood circulation and, most importantly, it extends the residence time in serum compared with the oral route. To support this observation, the *t*_1/2_ almost doubled and *k_el_* halved upon pulmonary administration (Table 3). This finding can be ascribed to a reservoir effect of the donor compartment that allows sustained diffusion of SA from the lungs to the blood circulation. This assumption also explains the high *t*_1/2_ after oral administration in the lungs, which surpasses that of the ASA DPI (Table 3). On the other hand, the ASA DPI suspension showed a fivefold higher clearance, ascribable to fast absorption as well as different deposition pattern that may favor mucociliary and macrophages scavenging [52]. In fact, dry powder inhalation can elicit higher penetration into the deep lungs, overcoming, in some cases, the mucociliary escalator system that is characteristic of the bronchial area where conventional liquid aerosols preferentially impact. Naturally, such considerations cannot be generalized as modern technologies can grant respiratory lung area penetration of liquid aerosols as well. However, this partial superiority of dry powders over liquid formulations, in our case, may explain the considerably higher relative bioavailability of ASA DPI (Table 3). These observations, along with the higher-than-expected AUC values, suggest that ASA DPI could help reduce dosages while, at the same time, grant similar effects of the conventional oral products, potentiating its action when local activity is sought. Moreover, an extended activity may be expected considering the longer residence in blood and lungs, as previously mentioned.

Overall, our findings apparently confirm the first and only report in humans with an ASA inhalable powder that pointed out an increased ASA bioavailability and faster antiplatelet activity onset compared with conventional oral products [16]. Such enhanced ASA diffusion between the lung and blood compartments was also observed in this work for SA, with the possibility of potentially reducing the required dose and dose frequency, thus envisaging therapeutic efficacy enhancement through ASA DPI. Beyond further quality assessment of the DPI formulation, successive studies will focus on the in vivo ASA/SA turnover and on the evaluation of the efficacy in models of inflammation and infection.

## 4. Conclusions

In this work, a novel high-performing ASA DPI for inhalation was successfully developed. The obtained formulation showed unprecedented features in term of respirability that may ensure high penetration in the deep lung. Albeit initial and with limitations, this study highlighted an improved PK profile for ASA DPI compared with oral administration. Such features attest to a potential enhanced efficacy of the pulmonary treatment when either local or systemic therapeutic effects are sought. Future works will explore such potentiality in the treatment of pulmonary inflammatory and infectious diseases.

## Figures and Tables

**Figure 1 pharmaceutics-14-02819-f001:**
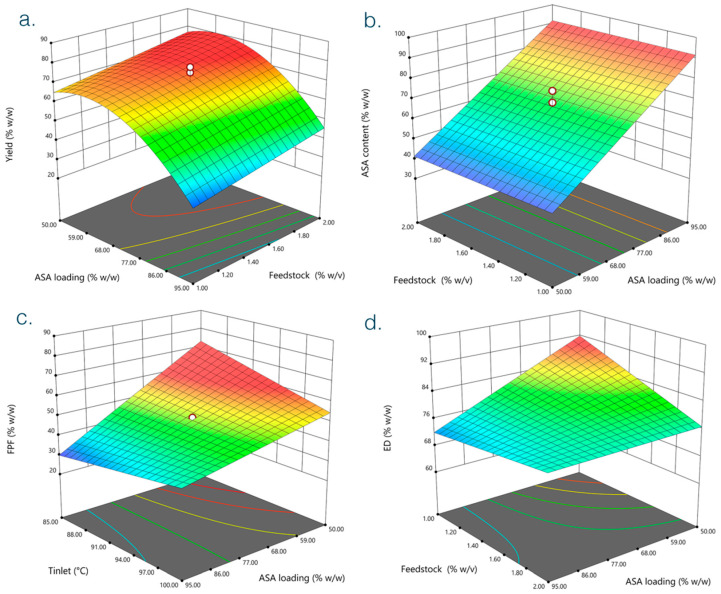
Surface plots of the four responses in Table 2: (**a**) Yield, (**b**) ASA content, (**c**) FPF, and (**d**) ED. Graphs were built against significant factors according to the results of ANOVA validation of the models obtained (see Appendix A). Color map: red-shaded = high, green-shaded = medium, and blue-shaded = low response values.

**Figure 2 pharmaceutics-14-02819-f002:**
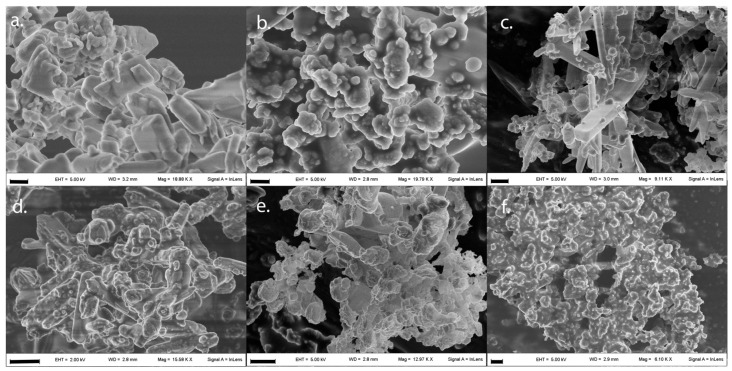
Morphological observation of samples produced at 95% of ASA loading. Pictures are representative of run# (**a**) 11, (**b**) 13, (**c**) 14, (**d**) 18, (**e**) 19, and (**f**) 20 of the experimental design in Table 2. Dimensional bars correspond to 2 µm.

**Figure 3 pharmaceutics-14-02819-f003:**
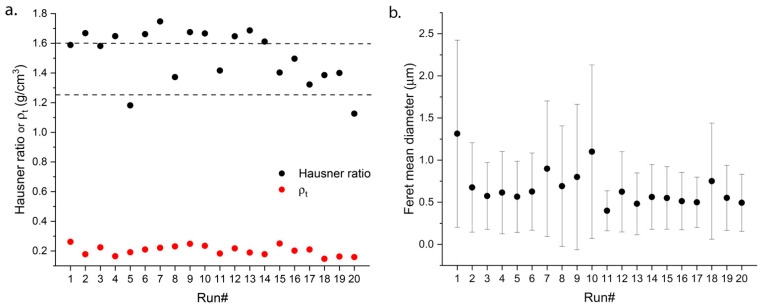
Theoretical flowability expressed as (**a**) H together with ρ_t_ and size distribution expressed as (**b**) FMD ± S.D. across the different runs of the experimental design. Horizontal lines delimit flowability regions: flowing (H < 1.25), partially flowing (1.25 < H < 1.6), and non-flowing (H > 1.6) powders.

**Figure 4 pharmaceutics-14-02819-f004:**
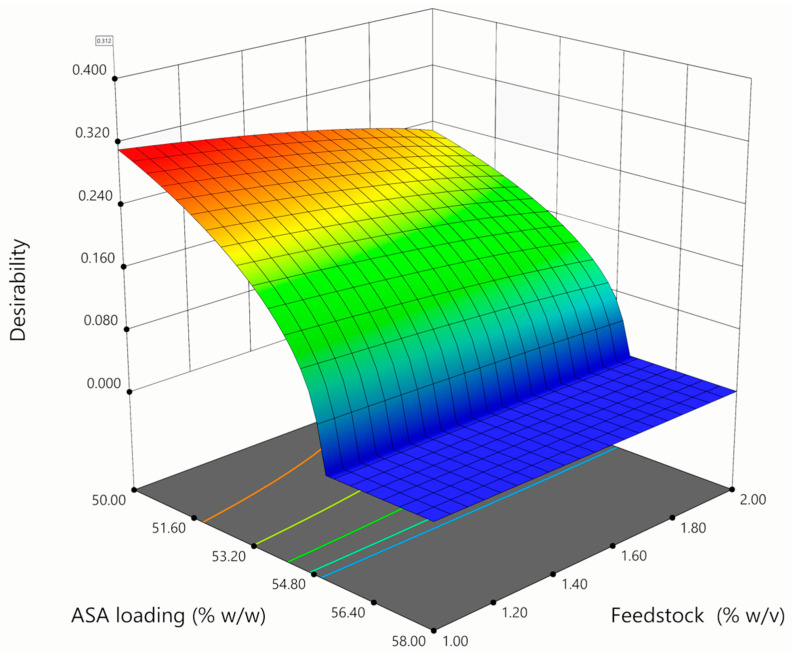
Desirability plot indicating the area of satisfaction (red-shaded) of the defined target properties of the obtained ASA inhalable powders: process yield ≥ 50%, ED ≥ 90%, FPF ≥ 50%, and ASA content ≥ 40%. Satisfaction of FPF and ED criteria was prioritized over the other responses. Maximum desirability was obtained at T_inlet_ = 85 °C.

**Figure 5 pharmaceutics-14-02819-f005:**
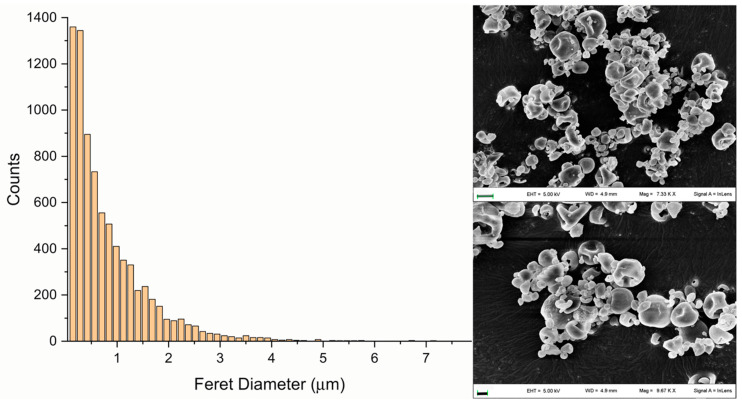
Feret’s size distribution (**left**) and morphology (**right**) of the chosen formulation corresponding to the run# 7 of the experimental design (Table 2). (**Right**) Dimensional bars correspond to 2 µm (**above**) and 1 µm (**below**).

**Figure 6 pharmaceutics-14-02819-f006:**
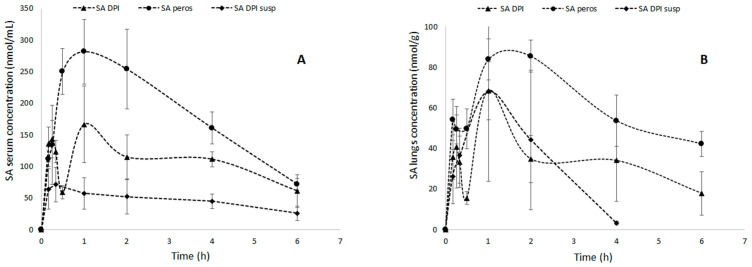
Serum (**A**) and lungs (**B**) SA time profiles after insufflation of ASA DPI (▲) and nebulization of ASA DPI suspension (♦) and after oral administration of ASA saturated solution (●). Data (n = 5) are expressed as mean ± SD.

**Table 1 pharmaceutics-14-02819-t001:** The 2^6−2^ fractional factorial design factors and levels.

	Factor	Low Level	High Level
A	ASA loading (%*w/w*)	50	95
B	Feedstock (%*w/v*)	1	2
C	Ethanol (%*v/v*))	30	40
D	Inlet temperature (°C)	85	100
E	Air flow rate (L/min)	95	110
F	Back pressure (mbar)	50	65

**Table 2 pharmaceutics-14-02819-t002:** Factors, levels, and response values of the 20 runs performed according to the 2^6−2^ fractional factorial design.

Factor	A	B	C	D	E	F	Responses			
	ASA Loading	Feedstock	Ethanol	T_inlet_	Air Flow Rate	Back Pressure	Yield	ASA Content	FPF	ED
Run#	% *w*/*w*	% *w*/*v*	% *v*/*v*	°C	L/min	mbar	% *w*/*w*	% *w*/*w*	% *w*/*w*	% *w*/*w*
1	50.0	2.0	40.0	85.0	95.0	50.0	79.8	53.6	64.9	77.0
2	95.0	1.0	30.0	85.0	110.0	50.0	37.1	94.7	27.4	71.4
3	50.0	1.0	30.0	100.0	95.0	65.0	63.4	39.9	58.4	98.7
4	95.0	2.0	40.0	100.0	110.0	65.0	40.4	93.0	39.3	83.7
5	50.0	2.0	40.0	100.0	95.0	65.0	72.6	45.3	51.4	73.6
6	50.0	2.0	30.0	100.0	110.0	50.0	73.8	39.3	55.0	79.7
7	50.0	1.0	30.0	85.0	95.0	50.0	76.7	42.9	85.8	98.5
8	72.5	1.5	35.0	92.5	102.5	57.5	59.8	62.7	50.8	77.5
9	50.0	1.0	40.0	85.0	110.0	65.0	65.7	36.7	76.5	103.2
10	50.0	2.0	30.0	85.0	110.0	65.0	79.5	36.0	68.7	82.0
11	95.0	1.0	30.0	100.0	110.0	65.0	30.5	88.0	50.1	84.0
12	72.5	1.5	35.0	92.5	102.5	57.5	74.9	68.5	45.3	69.8
13	95.0	1.0	40.0	100.0	95.0	50.0	25.7	81.6	36.0	69.6
14	95.0	2.0	30.0	100.0	95.0	50.0	47.3	89.9	37.1	74.9
15	72.5	1.5	35.0	92.5	102.5	57.5	77.6	74.3	40.4	72.8
16	50.0	1.0	40.0	100.0	110.0	50.0	57.3	35.6	56.4	72.4
17	72.5	1.5	35.0	92.5	102.5	57.5	68.9	65.5	49.3	76.8
18	95.0	2.0	30.0	85.0	95.0	65.0	56.1	95.7	34.5	75.7
19	95.0	2.0	40.0	85.0	110.0	50.0	42.2	91.9	33.8	76.4
20	95.0	1.0	40.0	85.0	95.0	65.0	40.0	95.6	27.0	68.1

**Table 3 pharmaceutics-14-02819-t003:** Estimated SA serum and lung PK parameters applying NCA sparse data analysis using Phoenix^®^ for all tested routes of ASA and ASA dry powder (ASA DPI) administration.

	Serum		
Parameter	Estimate (±SE)		
	ASA DPI Suspension	ASA DPI	Oral
k_el_ (h^−1^)	0.154	0.176	0.317
t_1/2_ (h)	4.49	3.93	2.19
t_max_ (h)	0.333	1	1
C_max_ (±SE) (nmol/mL)	71.0 (8.6)	166.5 (27.1)	281.7 (22.9)
AUC_last_ (±SE) (h nmol/mL)	281.9 (29.8)	644.1 (53.2)	1115 (50)
AUC_inf_ (h nmol/mL)	449.1	990.2	1342
AUC_% extrapolated_ *	37.2	34.9	16.9
Relative bioavailability (F) **	1.33	3.36	-
	LUNG		
Parameter	Estimate (±SE)		
	ASA DPI suspension	ASA DPI	Oral
k_el_ (h^−1^)	1.06	0.23	0.176
t_1/2_ (h)	0.654	3.02	3.93
t_max_ (h)	1	1	2
C_max_ (±SE) (nmol/mL)	68.2 (44.3)	68.6 (6.4)	85.6 (3.5)
AUC_last_ (±SE) (h nmol/mL)	146.0 (4.7)	206.9 (21.3)	374.6 (12.0)
AUC_inf_ (h nmol/mL)	149.0	284.5	614.6
AUC_% extrapolated_ *	2.0	27.3	39.0
Relative bioavailability (F) **	0.97	3.21	-
AUC_lung_/AUC_serum_ #	0.52	0.32	0.34

* Calculated using Equation (8); ** relative to oral, calculated using Equation (7); # AUC_last_ values; ASA DPI dose, 3.33 ± 1.67 μmol; ASA DPI suspension dose 5.0 ± 0.178 μmol; oral ASA dose,19.94 ± 1.67 μmol.

## Data Availability

The data presented in this study are available on request from the corresponding authors.

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
