# Peer review of "Development and Pharmacokinetics of a Novel Acetylsalicylic Acid Dry Powder for Pulmonary Administration"

_pharmaceutics, 2022, doi:10.3390/pharmaceutics14122819_

Round 1
Reviewer 1 Report
The manuscript presents the results of a sound complex study on an important topic. To enforce the presentation, more references can be discussed in the introduction in relation to using the nanoparticles of drugs in the formulations with excipients as inhalation powders or as aerosols. As examples I can suggest to look through the last issues of this journal, many of which were dedicated to these topics.
Pharmaceutics | Sections (mdpi.com)
I can also suggest to look at the papers:
Tolstikova, T.G. et al (2012). Research and Development of a New Safe Form of Drugs, In: Ruzer, L. S., & Harley, N. H. (Eds.). Aerosols handbook: measurement, dosimetry, and health effects. CRC press, pp. 249-284
Ogienko A.G. et al (2017). Large porous particles for respiratory drug delivery. Glycine-based formulations, European Journal of Pharmaceutical Sciences, 110, 148-156.
Onischuk A.A. et al (2014). Generation, inhalation delivery and anti-hypertensive effect of nisoldipine nanoaerosol, Journal of Aerosol Science, 78, 41-54.
Onischuk A.A. et al (2009). Analgesic effect from ibuprofen nanoparticles inhaled by male mice, Journal of Aerosol Medicine and Pulmonary Drug Delivery, 22, 3, 245-253.
Onischuk A.A. et al (2008). Anti-inflammatory effect from indomethacin nanoparticles inhaled by male mice, Journal of Aerosol Medicine and Pulmonary Drug Delivery, 21, 3, 231-243.
An’kov, S. V. et al (2016). Analgesic Effect of Several Nonsteroidal Anti-Inflammatory Drug Nanoaerosols. Pharmaceutical Chemistry Journal, 49(10), 680-682.
In the Abstract I would suggest to write not vaguely: "through a design of experiment approach", but more clearly "using nano spray-drying technique and leucine as an excipient"
The authors report that "ASA-leucine dry powders were prepared by using the Buchi nano spray-dryer B90 103 (Büchi, Milan, Italy). Sodium acetylsalicylate and D-leucine". Why D-leucine, not L-leucine? Is a D-amino acid biologically acceptable in a medication?
In ref.23 correct the font (now all capitals).
Reviewer 2 Report
Pharmaceutics 2045948
Title: Development and pharmacokinetics of a novel acetylsalicylic acid dry powder for pulmonary administration
The study aimed to develop inhaled acetylsalicylic acid dry powder prepared by nano spray drying. The formulation and processing parameters were optimized using a fractional factorial design. The dry powder was characterized by its powder physical properties (e.g., morphology, particle size, flow ability) and aerodynamic properties. The pharmacokinetic study in rats of inhaled ASA dry powder, inhaled ASA suspensions were compared to oral administration. Overall, the manuscript is well written. The experiments were well-designed. The data is well-interpreted and supports the conclusions. This manuscript can be accepted after some revisions.
1. In section 2.6, what is the pressure drop across the handler at 60 L/min?
2. In section 2.7.2, please give more detail on how the author dosed the rats using a tracheal catheter and insufflator (e.g., the volume of air).
3. What is the rationale for choosing leucine as the main excipient?
4. Does the drug loading affect the crystallinity of the drug? Any XRD or DSC data?
5. In addition to drug loading, can you explain more how other formulation and processing parameters affect the aerosol performance?
6. Line 399-400, how did the author select the dose for pulmonary administration? Any references?
7. Please improve the quality of SEM figures.
